# Oral Administration of *Armillaria mellea* Mycelia Promotes Non-Rapid Eye Movement and Rapid Eye Movement Sleep in Rats

**DOI:** 10.3390/jof7050371

**Published:** 2021-05-10

**Authors:** I-Chen Li, Ting-Wei Lin, Tung-Yen Lee, Yun Lo, Yih-Min Jiang, Yu-Hsuan Kuo, Chin-Chu Chen, Fang-Chia Chang

**Affiliations:** 1Biotech Research Institute, Grape King Bio Ltd., Taoyuan 320, Taiwan; ichen.li@grapeking.com.tw (I.-C.L.); tingwei.lin@grapeking.com.tw (T.-W.L.); yihmin.jiang@grapeking.com.tw (Y.-M.J.); yuhsuan.kuo@grapeking.com.tw (Y.-H.K.); 2Department of Veterinary Medicine, School of Veterinary Medicine, National Taiwan University, Taipei 106, Taiwan; r09454004@ntu.edu.tw (T.-Y.L.); f05629004@ntu.edu.tw (Y.L.); 3Institute of Food Science and Technology, National Taiwan University, Taipei 106, Taiwan; 4Department of Bioscience Technology, Chung Yuan Christian University, Taoyuan 320, Taiwan; 5Department of Food Science, Nutrition and Nutraceutical Biotechnology, Shih Chien University, Taipei 104, Taiwan; 6Graduate Institute of Brain and Mind Sciences, College of Medicine, National Taiwan University, Taipei 106, Taiwan; 7Graduate Institute of Acupuncture Science, College of Chinese Medicine, China Medical University, Taichung 404, Taiwan

**Keywords:** *Armillaria mellea*, GABA, HPLC, sesquiterpenoids, insomnia

## Abstract

The present study aimed to explore whether water and ethanol extracts of *Armillaria mellea* mycelia produce sedative and hypnotic effects in rats. Male Sprague–Dawley rats were surgically implanted with two electroencephalogram electrodes on the skull and an electromyogram electrode on neck muscle to evaluate the alterations in rapid eye movement (REM) and non-REM (NREM) sleep after oral administration of the water and ethanol extracts. Following post-surgical recovery, thirty-six rats were randomly divided into four treatment groups and two control groups. They were treated orally with vehicle, 75 and 150 mg/kg doses of water and ethanolic extracts 15 min prior to the onset of dark (active) period. Electroencephalography results showed that the low dose of *A. mellea* mycelia water extract increased REM sleep time while the high dose enhanced both REM and NREM sleep times during the subsequent light (rest) period. On the other hand, although the low dose of *A. mellea* mycelia ethanolic extract did not alter both NREM sleep and REM sleep during the dark and light periods, the high dose increased both REM and NREM sleep during the light periods in naive rats. The HPLC-DAD analyses of both extracts allowed the identification of GABA and seven sesquiterpenoids. Based on these findings, the present study showed for the first time that water and ethanolic extracts of *A. mellea* mycelia, containing a source of biologically active compounds, could increase both NREM sleep and REM sleep during the rest period and may be useful for the treatment of insomnia.

## 1. Introduction

Sleep is essential in sustaining healthy physical and mental health. It has important effects in many vital physiologic functions, such as development, energy conservation, brain waste clearance, cognitive support, hormonal balance and immune function modulation [1]. Sleep occurs in four stages that include non-rapid eye-movement (NREM) sleep (N1 to N3) and rapid eye-movement (REM) sleep, which constitutes 75 and 25 percent, respectively, of total time spent in sleep [2]. Throughout sleep, NREM sleep and REM sleep alternate 4–5 times in a cyclical fashion and are indicated by the different brain waves [2]. REM, also called paradoxical sleep, is characterized by intense dreams with desynchronized awake-like brain waves, while NREM is associated with deep sleep indicated by synchronized delta waves [2]. In a normal sleep architecture, both REM and NREM sleep are important in maintaining physical and mental homeostasis and disruption of these sleep stages has been known to associate with adverse health consequences, such as increased stress responses, mood disorders, memory deficits and eventually reduced quality of life [3]. For cumulative long-term effects of sleep loss, it increases the chance of developing various diseases, e.g., hypertension, diabetes, obesity, heart attack, stroke and depression [4].

Although National Sleep Foundation advises that adults aged 18–60 years are recommended to sleep at least 7 h every night [5], a survey conducted in 2014 by Centers for Disease Control and Prevention (CDC) showed that only 65% of adults reported a healthy duration of sleep while more than 35% among 444,306 adult respondents reported getting less than seven hours of sleep in a 24 h period. As the number of U.S. adults sleeping less than 7 h in a 24 h period increased from 38.6 million to 83.6 million between 1985 and 2014 [6,7], insufficient sleep has now been recognized as a “public health epidemic” according to the CDC. To improve sleep quality, most people utilize prescription medications that include benzodiazepines and non-benzodiazepines [8]. However, these drugs do not mimic physiological sleep [9] and long-term use often results in neurocognitive dysfunction, dependence and tolerance [10,11], leading to withdrawal syndromes after their decrease or discontinuation. Therefore, it is necessary to search for a novel hypnotic agent that exhibits a similar efficacy with fewer side effects as there is a growing interest among patients in alternative remedies for sleep disturbances.

*Armillaria mellea*, also known as honey mushroom, belongs to an edible and medicinal mushroom of the *Tricholomataceae* family and has a strong symbiotic relationship with orchid species *Gastrodia elata*, a common herb used in traditional Chinese Medicine [12]. In our previous studies, N⁶-(4-hydroxybenzyl) adenine riboside (NHBA), isolated from *G. elata*, produces significant sedative and hypnotic effects via increasing activity of GABAergic neurons in the ventrolateral preoptic area [13]. As *G. elata* requires *A. mellea* for growth and survival [14], it is therefore suspected that NHBA of *G. elata* may be a metabolite from the absorbed nutrients of *A. mellea*. Moreover, it was reported that *A. mellea* could be used in place of *G. elata* to treat various neurological conditions as it mirrors medical properties of *G. elata* with a half effective dosage of *G. elata* [15]. However, to date, there is no pharmacological evidence for the sedative-hypnotic effect of *A. mellea*. Hence, with the present study, we aimed to evaluate the sedative and hypnotic effects of *A. mellea* extracts and identify its major compounds using high-performance liquid chromatography with a photodiode array detector (HPLC-DAD).

## 2. Materials and Methods

### 2.1. Sample Preparation, Proximate Composition and Extraction

*Armillaria mellea* (# 36361) was purchased from the Bioresource Collection and Research Center (BCRC) of Taiwan and maintained on Potato Dextrose Agar (PDA) slants at 25 °C. After 10 days, 1 cm^3^
*A. mellea* was removed from the PDA and transferred to a 2 L flask containing 1 L synthetic culture medium (composed of 2% glucose, 1% soybean powder, 0.1% yeast extract and 0.1% peptone, adjusted to pH 4) at 25 °C for 10 days with shaking at 120 rpm. This fermentation process was then scaled up from a 2-L shake flask to 200 L and 20 ton fermenters, for 7 days and 10 days, respectively. After fermentation with controlled pH, agitation and temperature, the whole broth was harvested, lyophilized, grounded to a powder and stored in a desiccator at room temperature.

For aqueous and ethanolic extract preparation, the freeze-dried *A. mellea* powder was suspended at 1:20 *w*/*v* in water and ethanol (95%). Briefly, the mixture of the aqueous suspension was boiled at 121 °C for 15 min while the mixture of the ethanolic suspension was sonicated in bath sonicator for 1 h, Afterward, both suspensions were passed through Whatman filter paper NO.4 and concentrated through a rotary evaporator (R-220, Büchi Labortechnik AG, Switzerland). Dried aqueous and ethanolic extracts were then further reconstituted in water and ethanol, respectively, to obtain a final concentration of 100 mg/mL, which were filtered through a sterilized 0.22 μm syringe and stored at 4 °C before exposure to the animals.

### 2.2. Animals, EEG Acquisition and Analysis

The protocol for animal experiments was approved by the Institutional Animal Care and Use Committee (IACUC approval: NTU107-EL-00182) of National Taiwan University. Eight-week-old male Sprague Dawley (SD) rats weighing 250–300 g obtained from BioLASCO Taiwan Co., Ltd (Taipei, Taiwan). were kept individually in recording plastic cage, in which the temperature is maintained at 23 ± 1 °C with a 12 h light/dark cycle and atmospheric humidity of 50–60%. Food and water were available ad libitum. After an adaptation period (7 days), stereotaxic surgery was employed to implant EEG and EMG electrodes and a microinjection cannula and the rats were used for polysomnographic recordings.

Male Sprague–Dawley rats were anesthetized with Zoletil (50 mg/kg; Virbac, Carros, France) and xylazine (14.8 mg/kg; Sigma-Aldrich, MO, USA) before surgically implanted with EEG screw electrodes and EMG electrodes. Three EEG screw electrodes were implanted in the frontal and parietal lobes of the right hemisphere and the occipital lobe of the left hemisphere while two EMG electrodes were implanted in the neck muscle. Signals from the EEG and EMG electrodes were fed into an amplifier (Colbourn Instruments, Lehigh Valley, PA, USA; model V75-01), filtered between 0.1 and 40 Hz and digitized at a sampling rate of 128 Hz (NI PCI-6033E; National Instruments, Austin, TX, USA). For sleep-wake activity, NREM sleep is characterized by large-amplitude EEG slow waves, high power density values in the delta frequency band (0.5–4.0 Hz), a relaxed muscle tone from EMGs and lack of gross body movements. During REM sleep, the amplitude of the EEG is reduced, the predominant EEG power density occurs within the theta frequency (6.0–9.0 Hz), the EMGs exhibit muscle atonia with low EMG amplitudes and there are phasic body twitches. During waking, the rats are generally active with robust EMG amplitudes and there are protracted body movements.

### 2.3. Pharmacological Treatment

Following 7 days postsurgical recovery, rats were randomly divided into four treatment groups and two control groups, each consisting of six rats. Aqueous extract and ethanolic extract of *A. mellea* were suspended in distilled water and 5.5% ethanol (vehicles), respectively. Two vehicles, two doses (75 and 150 mg/kg) of *A. mellea* aqueous extract and two doses (75 and 150 mg/kg) of *A. mellea* ethanolic extract were administered orally 15 min before the onset of dark period. Since rats are nocturnal animal with less sleepiness during the dark (active) period and higher sleepiness during the light (rest) period, administration of substance prior to the dark period may more easily to observe the hypnotic effect during dark period. Nevertheless, the effective time was observed during the subsequent light period (refer to the Results section), this may be due to the pharmacokinetics of these extracts. The sleep-wake activity was recorded for 24 h (hrs). After 24 h, the recorded signals were segmented in 12-s epochs to visually identify vigilance states (wakefulness, NREM sleep, REM sleep) based on previously defined criteria [16].

### 2.4. HPLC-DAD Analysis of Chemicals from the Extracts

Compounds mellendonal B (Am-Q), mellendonal C (Am-E), armillane (Am-T), melleolide Q (Am-R), 6′-Chloro-5′-methoxy-armillane (Am-V), armillarikin (Am-P) and armillaridin (Am-O) were gifted by Professor Chien-Chih Chen at National Research Institute of Chinese Medicine. These compounds were determined using a Hitachi L-5000 series LC system (Hitachi Corporation, Tokyo, Japan) equipped with a L-5110 pump, L-5310 column oven, L-5260 autosampler, L-5430 diode array detector and EZChrom Elite software. The analytical column was a Kinetex C18 100A column (5 µm, 150 × 4.6 mm, Phenomenex, Torrance, CA, USA) maintained at 40 °C. The mobile phases consisted of 0.1% formic acid in water (A) and acetonitrile (B). The gradient was as follows: 0 min, 5% B, 0–3 min, 5–40% B, 3–13 min, 40–60% B, 13–20 min, 60–100% B, 20–25 min, 100% B, 25–26 min, 100–12% B, 26–30 min, 12% B, which was pumped at a flow rate of 1.0 mL/min, injected volume of 10 μL and detection wavelength of 254 nm.

### 2.5. HPLC-DAD Analysis of γ-Aminobutyric Acid (GABA) in the Extracts

Standard of GABA was procured from Sigma-Aldrich, Louis, MO, USA and detected using an Agilent 1260 HPLC system (Agilent Technologies, Palo Alto, Santa Clara, CA, USA) consisted of a G7111B quaternary pump with an in-line 4-channel vacuum degasser, a G7129A autosampler, a G7116A thermostatic column oven and a G7115A photodiode array detector. GABA was online-derivatized with the O-phthaldialdehyde and 9-fluorenylmethylchloroformate. The analytical column was an Ailgent AdvanceBio AAA column (2.7μm, 4.6 mm × 10 mm, Agilent Technologies, Palo Alto, Santa Clara, CA, USA) with detection at 338 nm, 4 nm (reference = 390 nm, 20 nm). Mobile phase A was 10 mM disodium phosphate +10 mM sodium tetraborate (pH 8.2) and phase B was acetonitrile–methanol–water (45:45:10, v/v/v). The flow rate was 1.5 mL/min with a gradient condition as follows: 0–19.5 min, 7–90% B, 19.5–23 min, 90–7% B. Agilent ChemStation software was used for instrumental control and data acquisition. The identification of GABA in the samples was carried out by comparison with the retention times of the standards.

### 2.6. Statistical Analysis

All data were expressed as the mean ± standard error of the mean (SEM). The statistical significance of the amounts of NREM sleep, REM sleep and wake, were analyzed by the use of one-way analyses of variance (ANOVA). In all cases, *p* < 0.05 was taken as the level of significance. The alteration during the 12 h dark period or light period was statistically analyzed and when it reached statistically significant change, the particular time period involved in the alteration was further compared.

## 3. Results

### 3.1. The Effects of Water-Extracted A. Mellea on Spontaneous Sleep-Wake Activity

Results indicated that NREM sleep was not altered during both the 12 h dark (active) period and 12 h light (rest) period; however, REM sleep was significantly increased during the 12 h light period, especially the first five hours (post-injection hours 13–17) of the light period, when rats received 75 mg/kg water-extracted *A. mellea* (Figure 1A). The percentage of time spent in REM sleep increased from 12.08 ± 0.91% obtained from vehicle to 15.89 ± 1.29% during the 12 h light period and it increased from 11.73 ± 1.54% to 18.98 ± 1.72% during hours 13–17 (Figure 1B; *n* = 6, *p* < 0.05). When the high dose (150 mg/kg) water-extracted *A. mellea* was given before the dark period, NREM sleep significantly increased during the 12 h light period, especially the hours between 18–20 (Figure 1A). The percentage of time spent in NREM sleep increased from 50.07 ± 1.90% obtained after vehicle to 58.82 ± 1.87% and it increased from 48.40 ± 3.03% to 64.50 ± 2.22% during hours 18–20 (Figure 1B; *n* = 6, *p* < 0.05). Moreover, REM sleep was enhanced during hours 10–12 of the dark period and the first 8 h of the light period (Figure 1A). The percentage of time spent in REM sleep increased from 11.43 ± 1.36% to 17.06 ± 1.16% during hours 10–20 (Figure 1B; *n* = 6, *p* < 0.05). The change in wakefulness was a mirror effect in response to both NREM sleep and REM sleep alterations. Wakefulness during the 12 h light period was reduced from 34.85 ± 2.47% obtained after vehicle to 26.30 ± 2.55%, when high dose water extract *A. mellea* was given before the dark period (Figure 1A).

### 3.2. The Effects of Ethanol-Extracted A. Mellea on Spontaneous Sleep-Wake Activity

Although 75 mg/kg ethanol-extracted *A. mellea* has no effect on both spontaneous NREM sleep and REM sleep during both the dark period and the light period, the high dose (150 mg/kg) ethanol-extracted *A. mellea* significantly enhanced NREM sleep during the 12 h light period, especially the hours 19–21 during the light period (Figure 2A; *n* = 6, *p* < 0.05). The percentage of time spent in NREM sleep during the 12 h light period increased from 41.32 ± 2.79% obtained after vehicle to 50.37 ± 2.29% and it also increased from 33.45 ± 4.93% to 50.54 ± 4.69% during hours of 19–21 (Figure 2B; *n* = 6, *p* < 0.05). Moreover, REM sleep also increased after administration of 150 mg/kg ethanol-extracted *A. mellea* during hour 15 and hour 20 of the light period, which the percentage of time spent in REM sleep increased from 14.04 ± 3.31% to 25.42 ± 3.04% and from 8.34 ± 3.59% to 25.00 ± 2.63%, respectively (Figure 2A; *n* = 6, *p* < 0.05). Wakefulness during the 12 h light period was reduced from 45.59 ± 3.63% obtained after vehicle to 34.28 ± 3.36%, when high dose water extract *A. mellea* was given before the dark period (Figure 2A).

### 3.3. Characteristics of the Compounds in Water and Ethanol Extracted A. Mellea

HPLC-DAD analyses were performed to determine for the first time the chemical profiles of ethanol and water extracted *A. Mellea* exhibiting remarkable biological activities. The representative LC chromatograms at 254 nm of these two extractions is shown in Figure 3A. The structures and quantification of all identified compounds are shown in Figure 3B,C, respectively.

### 3.4. Analysis of GABA in Water and Ethanol Extracted A. Mellea

Standard GABA, water extract and ethanol extracted of *A. Mellea* were analyzed under the same chromatographic conditions, with an identical retention time of 7.2 min as shown in Figure 4. GABA found in water and ethanol extracted *A. Mellea* was 337.2 µg/g and 0, respectively.

## 4. Discussion

Sleep is a basic physiological need as it removes cellular toxic byproducts which may have accumulated in the brain and allows the brain to function normally the next day [17]. With insomnia or sleep deprivation, it can cause poor performance in some executive functions and volume changes in cortical and subcortical gray matters, including the key areas involved in Alzheimer’s disease, as well as decreased white matter diffusivity [18]. According to the previous research, *A. mellea* shares a symbiotic relationship with *Gastrodia elata*, a medicinal herb in treating insomnia. Due to their symbiosis, *A. mellea* may possess pharmacological properties similar to *G.* elata. However, the sedative and hypnotic effects and the chemical constituents of *A. mellea* have not been studied. Therefore, the present study was the first to explore the sedative and hypnotic effects of *A. mellea* in rodents and evaluate the chemical profile of *A. mellea* extracts.

By analyzing high-density EEG and EMG recordings collected in Sprague-Dawley rats, sleep is broadly divided into REM and NREM [19]. The present study showed that both water and ethanol extracted *A. mellea* prolonged NREM and REM sleep time as compared to controls, with higher concentration more effective than lower concentration. Increased amount of NREM sleep, via an increase in delta waves, can impact brain development and promote optimal brain/psychological health [20]. In the latest studies, enhanced NREM sleep via pharmacological interventions could alleviate axonal damage and cognitive decline after rodent traumatic brain injury, potentially offering a new non-invasive treatment option [21]. Moreover, increased NREM sleep predicts better memory performance as disrupted hippocampal oscillatory activity during NREM sleep impairs memory consolidation [22,23]. Taken together, the findings of this study suggested that a diet rich in *A. mellea* could assist in increasing time spent in NREM sleep, leading to improved cognitive processes and memory consolidation.

Compared with NREM sleep, REM sleep is marked by intense brain activity and often associated with very vivid dreams. Evidence has suggested that REM sleep is associated with memory consolidation [24], hippocampal excitability [25] and emotions [26]. Despite the physiological functions of REM sleep remain unclear, a decreased percentage of REM sleep was associated with an increased risk of brain-related diseases and all-cause mortality [27]. A previous study has found that increased time in N1 sleep and less time in REM sleep were associated with worsening cognitive performance [28]. Moreover, people diagnosed with clinically significant depressive symptoms spent more time in N2 sleep and less time in REM sleep [29]. Additionally, in a parallel analysis conducted in 2 instrumental, well-characterized, population-based sleep cohorts, which consisted of more than 4000 patients from the Outcomes of Sleep Disorders in Older Men Sleep Study (*n* = 2675) and Wisconsin Sleep Cohort (*n* = 1386), showed that in every 5% reduction of REM sleep, the mortality rate among older and middle-aged adults increases by 13% to 17% [30]. As the rats eating *A. mellea*, compared to control diet, had increased time spent in REM sleep in this study, it may be concluded that *A. mellea* could improve neurobiological outcomes and decrease mortality rate among adults.

Since *A. mellea* exhibits remarkable sedative and hypnotic effects in this study, HPLC-DAD analyses were performed for the first time to determine the chemical profiles of ethanol and water extracts. Sesquiterpene aryl esters (sesquiterpenoids) are the main active components of *A. mellea*. Up to now, more than 71 natural sesquiterpenoids have been isolated from fruiting bodies and mycelium of *A. mellea* [31]. In line with this result, sesquiterpenoids were also found in the water and ethanol extracted *A. mellea* compositions of this study. Melledonal B and melledonal C were identified in both ethanol and water extracts while armillaridin, armillarikin, armillane, melleolide Q and 6′-Chloro-5′-methoxy-armillane were detected only in extract using ethanol, when compared with standard compounds.

Pharmacologically, the most relevant feature of sesquiterpenoids from *A. mellea* is their ability to exhibit antibacterial [32] and anticancer activities [33]. Lately, protoilludane sesquiterpenoid aromatic esters from *A. mellea* were found to possess a markedly antidepressant-like activity using the open field test, tail suspension test and forced swimming test in mice [34]. A great deal of evidence has suggested that depression and sleep disturbances have a bidirectional relationship and sleep problems would resolve as an associated symptom with the treatment of depression [35]. In the present study, the potential of these sesquiterpenoids to treat sleep disturbances was highlighted for the first time. However, as it is difficult to speculate which of these compounds is mostly responsible for the sedative-hypnotic effects, further research regarding the bioactivity of a single compound or a synergistic mix of compounds is needed before any conclusions can be made.

While the sesquiterpenoids in *A. mellea* may directly or indirectly regulate sleep, the contribution of neurotransmitters to induce sedative and hypnotic effects cannot be ruled out. In the brain, there are many endogenous neurotransmitters, including amino acids, monoamines and acetylcholine that are involved in the sleep-wake regulation [36]. Among these neurotransmitters, GABA is one of the best-known sleep-promoting agents and its receptor is a current target for the majority of sleep medications [37]. For the first time, this study showed that GABA was observed in the water extract and not detected in the ethanol extract of *A. mellea*, which may explain why the sedative-hypnotic effect was more effective with water extract than the ethanol extract under the same concentration. Although exogenous GABA administration has long thought to be unable to cross the blood–brain barrier (BBB), studies have reported that the oral GABA administration can act on the peripheral nervous system of the digestive organs and improves the sleep behaviors [38,39]. Moreover, there are studies which reported that GABA does cross BBB, even if it is a small amount [40,41]. The increase of REM sleep by low dose of A. mellea water extract maybe due to the other components of the extract, but not GABA. Based on these findings, it can be presumed that GABA from *A. mellea* could, at least in part, activate receptors in the enteric nervous system, sent signals via the vagus nerve and contributed to the sleep promotion effect. Additional studies, however, are warranted to confirm this assumption and need to be extended to other regions of the nervous system.

## 5. Conclusions

In conclusion, the present study demonstrated that both water and ethanol extracted *A. mellea* shows sleep-promoting effect in rats and such effect was more effective with water extract than the ethanol extract. GABA, melledonal B and melledonal C were identified as the major constituents in the water extract while melledonal B and melledonal C armillaridin, armillarikin, armillane, melleolide Q and 6′-Chloro-5′-methoxy-armillane were detected in extract using ethanol. To the best of our knowledge, this study is the first to demonstrate that oral consumption of *A. mellea* containing these active compounds can promote sleep behavior.

## Figures and Tables

**Figure 1 jof-07-00371-f001:**
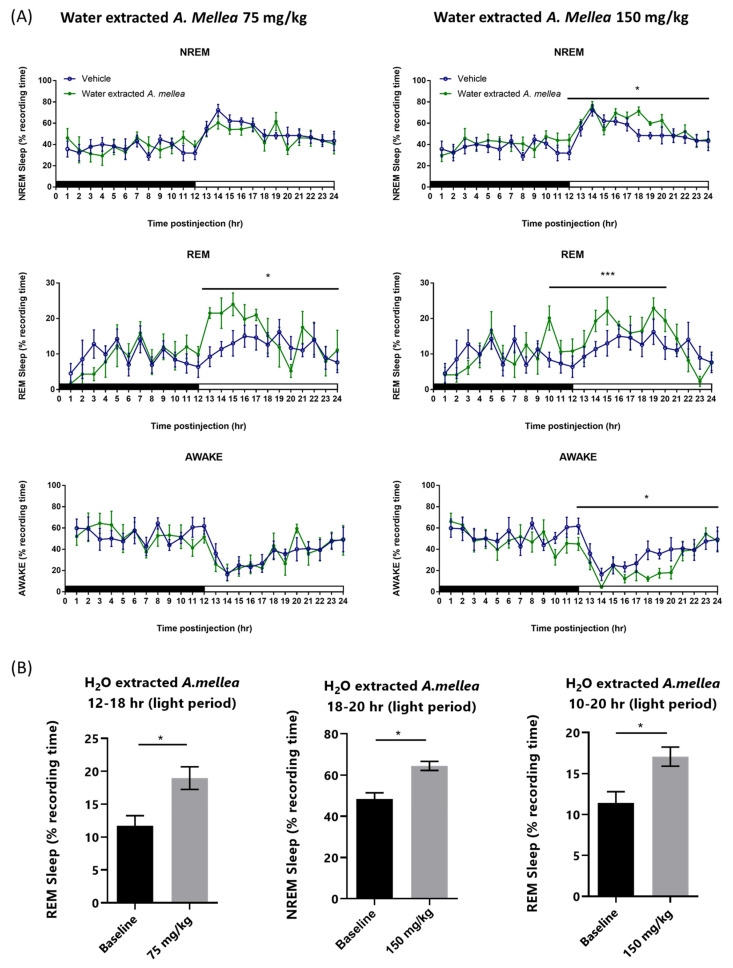
Water extracted *A. Mellea* increased NREM and REM in rats. (**A**) Time-course changes and (**B**) total amounts of REM, NREM sleep and awake produced after oral administration of water extracted *A. Mellea* at 75 mg/kg and 150 mg/kg. Open and closed circles stand for the profiles of vehicle and *A. Mellea* treatments, respectively. The *x*-axes indicate the 12 h dark (the black bar) and 12 h light periods (the white bar). Values are means ± SEM (*n* = 6). * *p* < 0.05 and *** *p* < 0.001 indicate significant differences compared with the vehicle group.

**Figure 2 jof-07-00371-f002:**
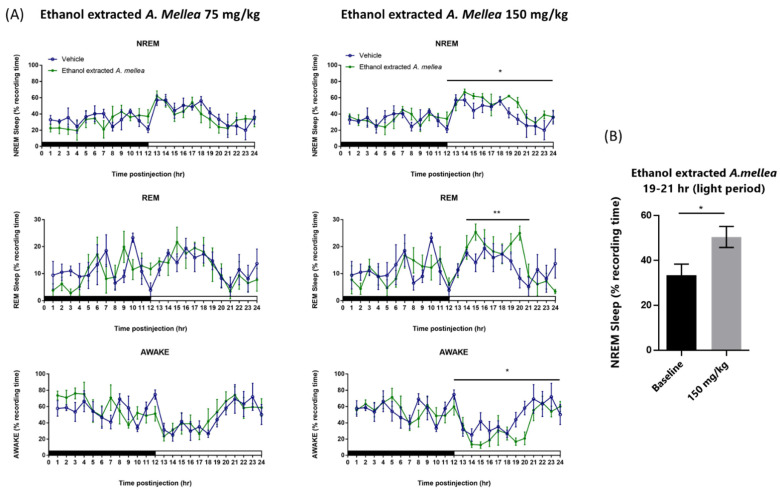
Ethanol extracted *A. Mellea* increased NREM and REM in rats. (**A**) Time-course changes and (**B**) total amounts of REM, NREM sleep and awake produced after oral administration of ethanol extracted *A. Mellea* at 75 mg/kg and 150 mg/kg. Open and closed circles stand for the profiles of vehicle and *A. Mellea* treatments, respectively. The *x*-axes indicate the 12 h dark (the black bar) and 12 h light periods (the white bar). Values are means ± SEM (*n* = 6). * *p* < 0.05 and ** *p* < 0.01 indicate significant differences compared with the vehicle group.

**Figure 3 jof-07-00371-f003:**
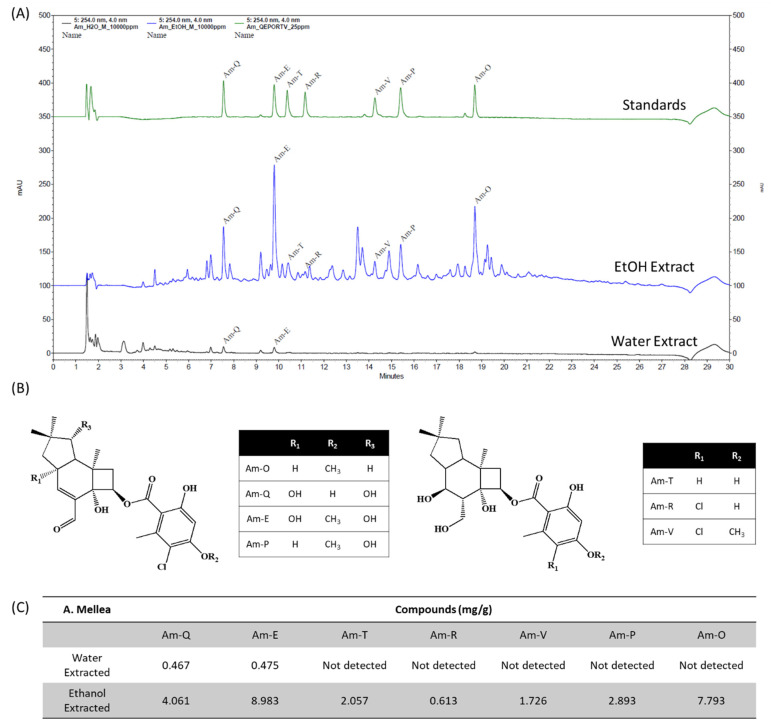
Quantification of compounds in water and ethanol extracted *A. Mellea*. (**A**) HPLC based peak chromatograms at 254 nm, (**B**) chemical structure of components and (**C**) quantification of identified compounds from ethanol and water extracts of *A. Mellea*. Am-Q: Melledonal B; Am-E: Melledonal C; Am-P: Armillarikin; Am-T: Armillane; Am-R: Melleolide Q; Am-V: 6′-Chloro-5′-methoxy-armillane; Am-O: Armillaridin.

**Figure 4 jof-07-00371-f004:**
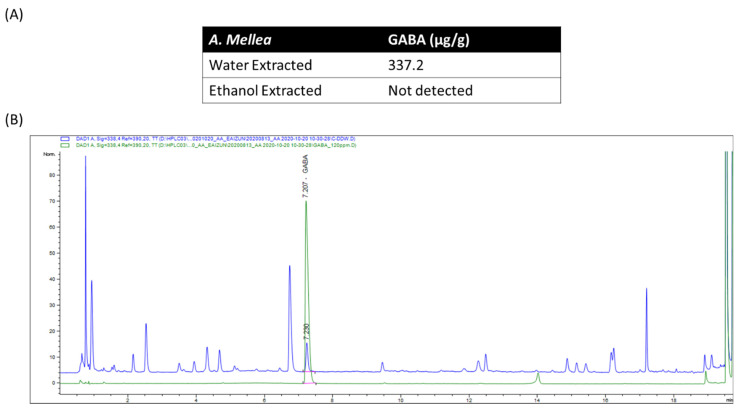
GABA quantification as evaluated by HPLC-DAD. (**A**) Concentration of GABA in water and ethanol extracted *A. mellea*. (**B**) Representative HPLC chromatograms of GABA standard sample (**bottom**) and the water extract of *A. mellea* mycelium (**top**) from 20-ton bioreactor (UV detection at 338 nm). Retention time of GABA was 7.2 min.

## Data Availability

The authors confirm that the data supporting the findings of this study are available within the article.

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
