# Peer review of "Oral Administration of *Armillaria mellea* Mycelia Promotes Non-Rapid Eye Movement and Rapid Eye Movement Sleep in Rats"

_jof, 2021, doi:10.3390/jof7050371_

Round 1
Reviewer 1 Report
This manuscript deals with an animal case and control experiment using male Sprague-Dawley rats to investigate the effect of A. mellea aqueous and ethanolic extracts on sleep architecture, which were administered orally 15 130 min before the onset of dark period. The results showed the low dose of A. mellea mycelia water extract increased REM sleep during the light period but the ethanolic extract did not show significant effect on sleep architecture. The high dose of A. mellea mycelia water and ethonolic extracts increased both REM and NREM sleep during the light period. The hypothesis was clear and the experimental design was robust. The results are also very interesting for a dose dependent effect on sleep architecture, in terms of low dose on REM sleep only but hight dose on both REM and NREM sleep. Only a few minor points needed to be addressed:
- The description "...microinjection cannula directly into the ventrolateral preoptic nucleus to evaluate the alterations in rapid eye movement (REM) and non-REM (NREM) sleep..." in the abstract seems confusing as the sleep architecture was NOT measured by microinjection cannula directly into the ventrolateral preoptic nucleus.
- The description "...the high dose increased both REM and NREM sleep during both the dark and light periods in naive rats..." in the abstract also seems confusing as there was no significant change during the dark period.
- Please give more explanation for giving the A. mellea mycelia water and ethonolic extracts before the dark period with a delayed effect on sleep architecture during the light period which is not commonly seen clinically for the use of hypnotics to treat insomnia disorder.
- As GABA, which was more related with enhancing NREM sleep, was found in water extract but not ethanol abstract, please give the rationale of increasing REM sleep by low dose of A. mellea mycelia water extract.
- Please also give the rationale for variable "time interval" for statistical comparisons in terms of 3, 6 and 10 hours. Could it be biased by type I error following multiple statistical comparisons?
Author Response
We thank reviewers’ comments. We respond reviewers’ comments point-to-point as follows.
Reviewer #1
This manuscript deals with an animal case and control experiment using male Sprague-Dawley rats to investigate the effect of A. mellea aqueous and ethanolic extracts on sleep architecture, which were administered orally 15 130 min before the onset of dark period. The results showed the low dose of A. mellea mycelia water extract increased REM sleep during the light period but the ethanolic extract did not show significant effect on sleep architecture. The high dose of A. mellea mycelia water and ethonolic extracts increased both REM and NREM sleep during the light period. The hypothesis was clear and the experimental design was robust. The results are also very interesting for a dose dependent effect on sleep architecture, in terms of low dose on REM sleep only but hight dose on both REM and NREM sleep. Only a few minor points needed to be addressed:
- The description "...microinjection cannula directly into the ventrolateral preoptic nucleus to evaluate the alterations in rapid eye movement (REM) and non-REM (NREM) sleep..." in the abstract seems confusing as the sleep architecture was NOT measured by microinjection cannula directly into the ventrolateral preoptic nucleus.
Response: We thank reviewer finding this mistake, and we changed the description as follows: “Male Sprague-Dawley rats were surgically implanted with two electroencephalogram electrodes on the skull and an electromyogram electrode on neck muscle to evaluate the alterations in rapid eye movement (REM) and non-REM (NREM) sleep after oral administration of the water and ethanol extracts.” on page 1, lines 19-22.
- The description "...the high dose increased both REM and NREM sleep during both the dark and light periods in naive rats..." in the abstract also seems confusing as there was no significant change during the dark period.
Response: We changed the description to “…..the high dose increased both REM and NREM sleep during the light periods in naive rats” on page 1, lines 29-30.
- Please give more explanation for giving the A. mellea mycelia water and ethonolic extracts before the dark period with a delayed effect on sleep architecture during the light period which is not commonly seen clinically for the use of hypnotics to treat insomnia disorder.
Response: We explained the rationale for giving extract prior to the dark period as follows:” Since rats are nocturnal animal with less sleepiness during the dark (active) period and higher sleepiness during the light (rest) period, administration of substance prior to the dark period may more easily to observe the hypnotic effect during dark period. Nevertheless, the effective time was observed during the subsequent light period (refer to the Results), this may be due to the pharmacokinetics of these extracts.” on page 3, lines 138-143.
- As GABA, which was more related with enhancing NREM sleep, was found in water extract but not ethanol abstract, please give the rationale of increasing REM sleep by low dose of A. mellea mycelia water extract.
Response: We explained the possible reason as follows “The increase of REM sleep by low dose of A. mellea water extract maybe due to the other components of the extract, but not GABA.” on page 9, lines 312-313.
- Please also give the rationale for variable "time interval" for statistical comparisons in terms of 3, 6 and 10 hours. Could it be biased by type I error following multiple statistical comparisons?
Response: We statistically analyzed the 12-h changes during the light period, when it reached statistically significant change, then the particular time period involved in the alterations will be further compared. This comparison avoids type 1 error in statistical analysis. The statement of “The alteration during the 12-h dark period or light period was statistically analyzed, and when it reached statistically significant change, the particular time period involved in the alteration was further compared” was added on page 4, lines 177-179.
Reviewer 2 Report
Comments to the authors:
The title should be changed to ‘Oral administration of Armillaria mellea mycelia promotes non-rapid eye movement and rapid eye movement sleep in rats.’
Fig 1.The ZT hours of REM seep and NREM sleep percentage are different; 75mg is for 6hr, 150 for 4hr, and then 10hr for REM sleep. It would be nice if here was a consistencies in the ZT hours and the duration.
This is a naïve question but the effects from extraction in the form of water and ethanol seem to be different. Would you explain the differences in these two extracts?
What is the change in wake from the results?
Author Response
We thank reviewers’ comments. We respond reviewers’ comments point-to-point as follows.
Reviewer #2
The title should be changed to ‘Oral administration of Armillaria mellea mycelia promotes non-rapid eye movement and rapid eye movement sleep in rats.’
Authors’ response: Thank you for the comments. We have changed the title as suggested in the revised manuscript (Page 1).
Fig 1.The ZT hours of REM seep and NREM sleep percentage are different; 75mg is for 6hr, 150 for 4hr, and then 10hr for REM sleep. It would be nice if here was a consistencies in the ZT hours and the duration.
Authors’ response: We thank reviewer’s comment. We statistically analyzed the 12-h changes during the light period, when it reached statistically significant change, then the particular time period involved in the alterations will be further compared. The statement of “The alteration during the 12-h dark period or light period was statistically analyzed, and when it reached statistically significant change, the particular time period involved in the alteration was further compared” was added on page 4, lines 177-179.
This is a naïve question but the effects from extraction in the form of water and ethanol seem to be different. Would you explain the differences in these two extracts?
Authors’ response: Thank you for the comments. Some of major differences between water and ethanol extract can be found in Figure 3. The ethanol extract contained major bioactive compounds such as Am-T, Am-R, Am-V, Am-P and Am-O while these bioactive compounds cannot be found in the water extract. Moreover, only GABA in water-extract but not in ethanolic extract (Fig 4).
What is the change in wake from the results?
Authors’ response: The alteration of wakefulness is a mirror effect in response to the summation of NREM sleep and REM sleep. In order to simply the representation of the results, we only demonstrated NREM sleep and REM sleep in this manuscript.
Round 2
Reviewer 2 Report
Oral administration of Armillaria mellea mycelia promotes NREM and REM sleep in rats.
Comments to the Authors:
- Thank you for the correct version of the title.
- Maybe I was not so clear with what I was asking. I see the logic behind why you decided to put only those figures; it would be great if you could provide the whole entire data set with the analysis at certain ZT hours so that it looks consistent and mention that the rest of the analysis set is included as supplementary.
- I was asking for Wake to be presented along with REM and NREM. Although wake did not show significant changes, it should be shown as a courtesy.
Overall, this study is interesting; however, it seems like the authors did not read between the lines. I would like to advise you to have the manuscript proofread as well.
Author Response
We thank reviewer’s comments. The responses to each comment are as follows.
Comments to the Authors:
- Thank you for the correct version of the title.
Response: We thank reviewer’s comment.
- Maybe I was not so clear with what I was asking. I see the logic behind why you decided to put only those figures; it would be great if you could provide the whole entire data set with the analysis at certain ZT hours so that it looks consistent and mention that the rest of the analysis set is included as supplementary.
Response: We provide the statistical analysis of the entire data during the 12h light period for NREM sleep, REM sleep and wakefulness after given either the water extract or ethanol extract (Figures 1 and 2). The statements are described on page 4, lines 178-179, lines 180-182, lines 183-186, lines 190-193; on page 5, lines 203-207, and lines 211-213.
- I was asking for Wake to be presented along with REM and NREM. Although wake did not show significant changes, it should be shown as a courtesy.
Response: The alteration of wakefulness is a mirror effect in response to the changes of NREM sleep and REM sleep. We provided the change of wakefulness in both Fig. 1 and Fig. 2, and the statements also could be found on page 4, lines 190-193, and page 5, lines 211-213.